# Identification of Torquetenovirus Species in Patients with Kawasaki Disease Using a Newly Developed Species-Specific PCR Method

**DOI:** 10.3390/ijms24108674

**Published:** 2023-05-12

**Authors:** Pietro Giorgio Spezia, Fabio Filippini, Yoshiro Nagao, Tetsuya Sano, Takafumi Ishida, Fabrizio Maggi

**Affiliations:** 1Department of Translational Research, University of Pisa, 56127 Pisa, Italy; 2Department of Paediatrics, Japan Community Health Care Organisation Osaka Hospital, Osaka 5530003, Japan; 3Department of Paediatrics, Fukuoka Tokushukai Hospital, Fukuoka 8160864, Japan; 4Department of Biological Sciences, University of Tokyo, Tokyo 1130033, Japan; 5Laboratory of Virology, National Institute for Infectious Diseases “Lazzaro Spallanzani”—IRCCS, 00149 Rome, Italy

**Keywords:** anellovirus, quantitative PCR, next-generation sequencing, primer set, quality score

## Abstract

A next-generation sequencing (NGS) study identified a very high viral load of Torquetenovirus (TTV) in KD patients. We aimed to evaluate the feasibility of a newly developed quantitative species-specific TTV-PCR (ssTTV-PCR) method to identify the etiology of KD. We applied ssTTV-PCR to samples collected from 11 KD patients and 22 matched control subjects who participated in our previous prospective study. We used the NGS dataset from the previous study to validate ssTTV-PCR. The TTV loads in whole blood and nasopharyngeal aspirates correlated highly (Spearman’s R = 0.8931, *p* < 0.0001, n = 33), supporting the validity of ssTTV-PCR. The ssTTV-PCR and NGS results were largely consistent. However, inconsistencies occurred when ssTTV-PCR was more sensitive than NGS, when the PCR primer sequences mismatched the viral sequences in the participants, and when the NGS quality score was low. Interpretation of NGS requires complex procedures. ssTTV-PCR is more sensitive than NGS but may fail to detect a fast-evolving TTV species. It would be prudent to update primer sets using NGS data. With this precaution, ssTTV-PCR can be used reliably in a future large-scale etiological study for KD.

## 1. Introduction

Kawasaki disease (KD) was first reported as a febrile illness characterized by mucocutaneous manifestations [1]. Subsequently, it was found that coronary aneurysms may develop in 40% of children with KD [2]. An increasing number of cases have been reported in Japan [3], as well as in other countries/regions [4,5]. However, the etiology of KD has not yet been determined.

Epidemiological studies implied an infectious (especially viral) etiology for KD [6,7,8]. Meanwhile, an increasing number of genotypes have been reported as predisposing factors for this illness [9,10,11,12]. Therefore, it is widely assumed that KD develops when a genetically predisposed individual is infected with a spectrum of microbes [13]. However, only a handful of prospective studies have explored associations between viruses and KD [14,15], and these pioneering prospective studies examined a limited number of viruses. Therefore, to identify associations between viruses and KD, we previously conducted a prospective study using comprehensive high-throughput next-generation sequencing (NGS) [16]. This pilot study identified very high viral loads of Torquetenovirus (TTV) only in KD patients and in none of the control individuals. However, it was estimated that a large sample size would be needed to identify a statistically significant association between TTV and KD. NGS sequencing remains costly and technically challenging to apply to such a large number of samples. As an alternative, PCR is used increasingly frequently to quantitate TTV viral load [17,18]. In particular, we have recently developed a species-specific PCR methodology that differentiates all TTV species. In this study, we aimed to examine the feasibility of, and to identify the problems in, applying this quantitative species-specific TTV-PCR (ssTTV-PCR) to an epidemiological study of KD.

## 2. Results

### 2.1. Quantification of Total TTV DNA in Individual Samples

First, we quantified total TTV DNA in whole blood (WB) and nasopharyngeal aspirate (NPA) from 11 KD patients and 22 control individuals who participated in our previous prospective study (Table 1) [16]. As shown in Figure 1, the TTV DNA loads in WB and NPA correlated highly (Spearman’s R = 0.8931, *p* < 0.0001, n = 33), which is consistent with a previous report [19]. Based on this finding, we used only WB DNA from each individual for the subsequent analyses.

### 2.2. High-Throughput NGS Sequencing and Bioinformatical Analysis

Livermore Metagenomics Analysis Tool Kit was applied to the high-throughput NGA datasets as described previously [16]. Briefly, a large number of viral reads were identified in the WB DNA which was pooled from 11 KD patients, and 99% of these reads were TTV7. In contrast, only a very small number of viral reads were found in the WB pooled from 11 diarrheal patients or from 11 children with respiratory infections. Similarly, 76% of the viral reads in the NPA pooled from 11 KD patients were TTV7. Rhinovirus A, respiratory syncytial virus, and human metapneumo virus were the most abundant viral species in the NPA obtained from 11 children with respiratory infections.

### 2.3. Selection of TTV Species Based on High-Throughput NGA Sequencing

The reads from pooled WB DNA from the KD, diarrhea control, and respiratory infection control groups were mapped to the reference sequences of TTV1–29 [20] and examined visually (examples are shown in Figure 2). Some of the TTV species were covered by reads throughout the entire genome: for example, TTV7 in the KD group (Figure 2a), TTV28 and TTV29 in the diarrheal control group (Figure 2b,c), and TTV22 in the respiratory infection control group (Figure 2d). These results suggested that these TTV viruses existed in the corresponding pooled samples. However, other TTV species were only covered partially, such as TTV8 and TTV19 in the diarrheal control group (Figure 2e,f), indicating that these TTV species did not exist in the corresponding pooled samples even though a large number of reads mapped to the non-coding region. By visual inspection of all 29 TTV species, we classified them as either present or absent. For subsequent analyses, we selected the TTV species that we identified as present in one or more of the three groups; thus, we chose TTV7, -1, and -24 present in the KD group (Figure 3a); TTV29, -13, -5, -24, -28, and -15 present in the diarrheal group (Figure 3b); and TTV5 and -22 in the respiratory infection group (Figure 3c).

### 2.4. ssTTV-PCR

We applied ssTTV-PCR to the 33 individual samples using the species-specific primer sets (Table 2). The viral loads in the individual samples are shown in Appendix A. In Table 3, we compare the presence of each TTV species identified by NGS sequencing to that identified by ssTTV-PCR. The ssTTV-PCR results were largely consistent with the pooled DNA NGS results. For most of the inconsistencies, ssTTV-PCR identified a TTV species at least in one individual of a group and NGS failed to detect the species in the pooled sample from the group. For example, NGS did not detect TTV15 in the respiratory infection control group, and NGS did not detect TTV22 in the diarrheal control group. These inconsistencies may have occurred because ssTTV-PCR may be more sensitive than high-throughput sequencing. However, in some situations, ssTTV-PCR did not detect a TTV species that was identified by NGS sequencing of pooled samples. For example, ssTTV-PCR did not detect TTV28 and TTV29 in any individuals in the diarrheal control group, whereas NGS detected TTV28 and TTV29 in the pooled sample from the diarrhea group. To determine the cause of this inconsistency, we re-examined the quality of the reads. For example, the quality score for the reads from the KD group mapped to the entire genome of TTV7 was high (Figure 4a), indicating that TTV7 unequivocally existed in the KD group. We then mapped the reads from the diarrheal control group to TTV28. Although a relatively large number of reads mapped to TTV28 (Figure 2b), the quality score was very low (Figure 4b), indicating that the visual impression was false and that TTV28 did not exist in the diarrheal group. We then examined why the ssTTV-PCR and NGS results were inconsistent for TTV29. The reads obtained from the diarrheal group were mapped to the reference sequence of TTV29 generating a consensus sequence with a high quality score (Figure 4 c). However, we found a substantial mismatch between this consensus sequence and the TTV29 reverse primer, which was based on the Genbank reference sequence (Figure 5), explaining why ssTTV-PCR did not identify TTV29 in any child in the diarrheal control group but NGS did. A single isolate of a TTV species can have >20% nucleotide divergence from the open reading frame (ORF) 1 capsid fragment [21]. The consensus sequence obtained in our study diverged from the reference sequence by 335/2200 bp (15.2%) for ORF1 alone. In addition, some isolates of a species may not amplify efficiently, especially if a primer mismatch is present at the 3′ end of the primer [22]. Because TTV28 and TTV29 exhibited inconsistent results between ssTTV-PCR and NGS, these TTV species were omitted from subsequent analyses.

### 2.5. Statistical Analysis

Table 4 shows the statistical association between the presence of each TTV species and the development of KD. Only TTV7 showed a marginal correlation with KD in our small study.

## 3. Discussion

Although it has been widely assumed that KD is of an infectious origin, the etiological agent has yet to be identified. During the global outbreak of COVID-19, SARS-CoV-2 was reported to cause a pediatric illness clinically similar to KD [23,24,25]. This news renewed interest in the possible viral involvement in KD. Previous modeling studies suggested that the etiological agent or agents of KD most likely establish a persistent infection [7,26]. In addition, the mean or median ages of patients with KD are younger than 3 years [27,28,29,30], implying that the main etiological agent for KD is highly infectious [31].

Our previous NGS analysis identified a high viral load of TTV, in particular TTV7, in KD patients [16]. TTV was first identified by Nishizawa et al., in 1997 [32] and assigned to a new family, Anelloviridae [20]. The pathogenicity of TTV is unknown except that it causes fever, respiratory symptoms, and elevation in hepatobiliary enzymes in children [33,34,35]. Most human beings are infected with at least one TTV species, establishing a persistent infection [36]. These characteristics of TTV are consistent with the properties predicted for the agent of KD. Therefore, we should consider TTVs as candidates for the etiology of KD.

Koch’s first postulate stated that the causative agent must be found in all the patients affected by the illness, but should not be found in healthy individuals. However, it was mathematically predicted that many children contract the etiological agent of KD, while only a small proportion of these infections develop KD [7]. Inevitably, therefore, Koch’s first postulate will not be fulfilled for KD. In addition, many viral species are not culturable, and viruses develop the illness in specific species. Further, experimental inoculation of a virus into a human individual is ethically impossible. Thus, Koch’s other postulates are not to be fulfilled as well. As a result, the exploration of KD etiology should start with epidemiological studies that statistically compare KD patients to control subjects. From a practical and ethical point of view, children with infectious illnesses constitute the control individuals. Considering the fact that KD cases often present as prodromal symptoms of an infection [37], a substantial proportion of control subjects should be positive with the KD agent. As a consequence, a study should contain a large number of KD patients and an equally large number of controls.

We compared high-throughput NGS sequencing to ssTTV-PCR and found that ssTTV-PCR is much less expensive and more sensitive than NGS. However, as this study showed, a primer set may not detect all the TTV variants, whereas NGS sequencing can detect the variants. This disadvantage of PCR has been reported previously [38]. Meanwhile, NGS results should also be interpreted with caution because a large number of reads mapped to the reference sequence of a virus does not necessarily confirm the presence of the virus. The quality scores of the mapped reads should be considered and should be fairly high throughout the sequences. In addition to the cost, this procedural complexity may make NGS less suitable for a large-scale epidemiological study.

TTV load has been established as a reliable indicator of immunosuppression and hence used in a large number of studies [18,19,39,40,41,42,43,44,45,46,47,48,49,50]. Therefore, ssTTV-PCR may be an optimal tool for epidemiological studies that explore the spectrum of TTV species in a large number of individuals. In the present study, ssTTV-PCR and NGS showed concordant results for most of the TTV species examined (i.e., TTV1, 5, 7, 13, 15, 22, and 24). However, ssTTV-PCR generated a false-negative result for TTV29, reflecting the great intra-species variation. Therefore, it would be prudent to compare ssTTV-PCR and NGS results for a select number of samples before launching a large-scale study.

We were aware that the sample size of this pilot study was too small to give a definitive answer to the question of KD etiology: instead, we proposed a reliable and affordable methodology for a future large-scale study. A future study should not only examine the relationship between TTV and the development of KD. The association between TTV and the clinical outcome of KD, most importantly cardiac complications, should also be an important research topic. Meanwhile, the development of cardiac lesions is closely associated with refractoriness to IVIG. However, the scoring models which were developed to predict IVIG resistance in Japan are not equally sensitive or specific in non-Japanese populations [51]. This fact implies that the relationship between complications of KD and the possible causative agents including TTV may also be dependent on the study population. Therefore, further efforts should be continued to explore the etiological agent(s) of KD, in diverse ethnicities, using up-to-date technologies including ssTTV-PCR.

## 4. Materials and Methods

### 4.1. Participants and Samples

We previously reported a prospective study that was conducted at the Japan Community Health Care Organisation (JCHO) Osaka Hospital [16]. The study complied with Japan’s ethical guideline [52] and was approved by the Ethics Committee of the JCHO Osaka Hospital. For each of the 11 KD patients who were enrolled between October 2015 and September 2016, a child with diarrhea and another child with a respiratory infection were assigned as controls; age, sex, and date of diagnosis were matched (Table 1). A child with one or more mucocutaneous symptoms resembling KD (e.g., rash, conjunctivitis) was not enrolled as a control subject because he/she may have incomplete KD. The guardians of all the participants provided written informed consent. From each of the 33 participants, WB and NPA were collected. NPA was prepared from one swab of nasal aspirate and another swab of pharyngeal aspirate, and both swabs were stored in a single tube of Universal Transfer Medium (UTM, COPAN, Brescia, Italy). DNA and RNA were extracted from WB and NPA as described previously [16]. The quantitative ssTTV-PCR in this study used only DNA samples. The viral loads in the NPA and WB were expressed as copies per microliter.

### 4.2. Quantification of Total TTV DNA in Individual Samples

Extracted viral DNA was amplified by real-time PCR amplification performed on a 7500 Fast Instrument (Applied Biosystems, Foster City, CA, USA). The TTV load was determined by a single-step TaqMan real-time PCR assay that targeted the highly conserved 5′UTR of the TTV genome, and the copy numbers were quantified as described previously [53,54]. Briefly, the method amplified a 63-nucleotide UTR fragment with high sensitivity (up to 10 viral genomes per milliliter of plasma/serum) and specificity.

### 4.3. NGS Sequencing of Pooled Samples

Using Nextera (Illumina, San Diego, CA, USA), a high-throughput NGS sequencer from Illumina (San Diego, CA, USA), we sequenced all the DNA and RNA contained in the pooled samples for each of the three groups: the KD, diarrheal control, and respiratory infection control groups [16]. We mapped these datasets (accession: DRA007000, DNA Data Bank of Japan) to the reference sequences of TTV species 1–29 (Figure 3).

### 4.4. Identification of TTV Species in Individual WB Samples by ssTTV-PCR

WB samples that were found to be positive by the UTR real-time PCR assay were amplified by nine PCR protocols that used species-specific primer sets for TTV species 1, 5, 7, 13, 15, 22, 24, 28, and 29. Briefly, rolling circle amplification was conducted in a total of 20 μL using an optimized mix containing ~10 ng of extracted DNA, 25 μM of exonuclease-resistant random primer, 4 mM of dNTPs, and 10 U of φ29 DNA polymerase. Amplification was performed at 30 °C for 17 h followed by inactivation of φ29 DNA polymerase at 65 °C for 10 min. The resulting linear double-stranded DNA products were quantified spectrophotometrically using a NanoDrop Lite (Thermo Fisher Scientific, Carlsbad, CA, USA), diluted 100-fold, and then used as the templates for ssTTV-PCR. All PCR assays followed standardized protocols with species-specific primers targeting open reading frame (ORF) 1 of the viral genome (Table 2). PCRs for TTV species 1, 5, 7, 13, 15, 22, and 24 were performed quantitatively. PCRs for TTV species 28 and 29 were qualitative. The sensitivity of each species-specific assay was tested and found to be ~100 copies per milliliter of WB. We sequenced the amplicons from the ssTTV-PCR by Sanger’s method to confirm that the amplicon was consistent with the reference species.

### 4.5. Statistical Analysis of the Association between Each TTV and KD

To investigate the association between each TTV species and KD, we employed exact logistic regression analysis for matched samples with a case-to-control ratio of 1:2. The explanatory variable was the presence or absence of a TTV species in WB identified by ssTTV-PCR. Stata 11 SE (StataCorp, College Station, TX, USA) was used for the statistical analysis.

### 4.6. Bioinformatical Analysis

We used CLC bio software (Qiagen, Hilden, Germany) for bioinformatical analyses.

## 5. Conclusions

We evaluated whether ssTTV-PCR can be used to explore the relationship between TTV species and KD. We did not intend to give a definitive answer to the etiological question of KD. Instead, this preliminary study evaluated the strengths and weaknesses of this newly developed tool. ssTTV-PCR is more sensitive and less costly than NGS. However, due to the mutations in the primer sites, ssTTV-PCR may fail to identify an evolving TTV species. Therefore, a future study which examines the role of TTV in the pathogenesis of KD should combine the benefit of the PCR-based methodology and that of NGS.

## Figures and Tables

**Figure 1 ijms-24-08674-f001:**
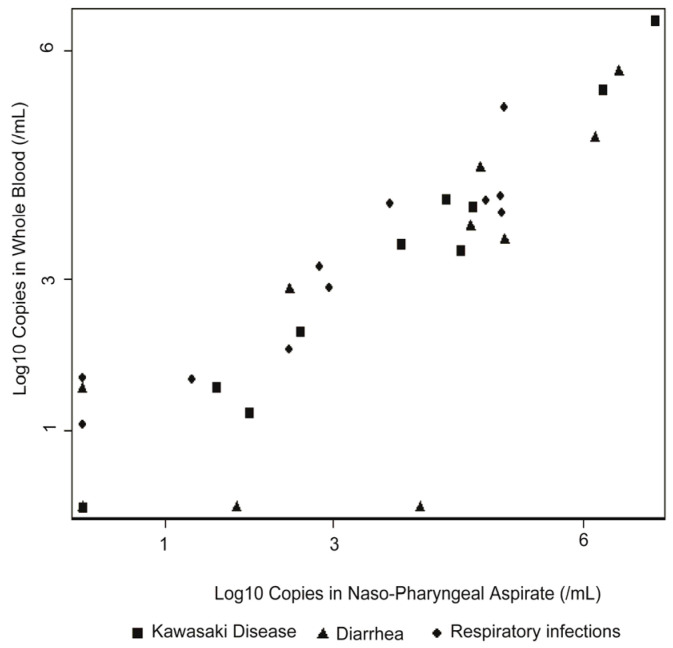
TTV viral loads in WB and NPAs. Total TTV loads in WB (Log10 copies/μL) and NPAs (Log10 copies/μL) were regressed. A log–log graph is presented. The viral loads were added with a value of 1 before being transformed into a logarithm. After addition of 1, four data points overlapped at Log_10_(X) = 0 and Log_10_(Y) = 0.

**Figure 2 ijms-24-08674-f002:**
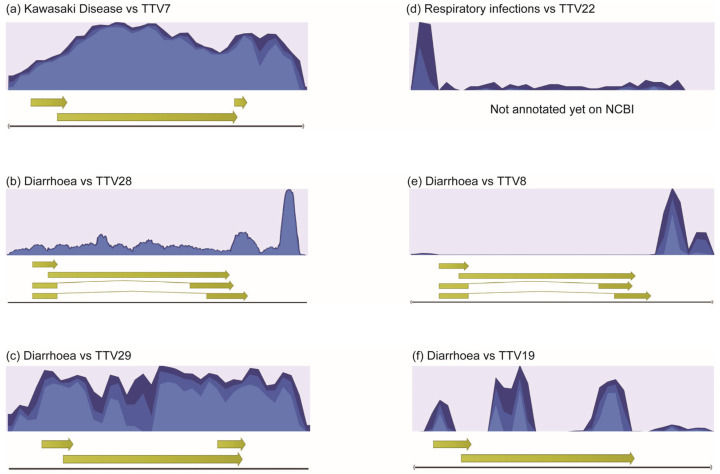
Coverage of high-throughput sequencing reads. The coverages (or depths) of reads along the genome of each TTV species were visualized. (**a**) Reads pooled from the KD group were mapped to the TTV7 reference sequence. (**b**) Reads from the diarrhea control group mapped to TTV28. (**c**) Reads from the diarrhea control group mapped to TTV29. (**d**) Reads from the respiratory infection control group mapped to TTV22. (**e**) Reads from the diarrhea control group mapped to TTV8. (**f**) Reads from the diarrhea control group mapped to TTV19. Yellow arrows represent open reading frames.

**Figure 3 ijms-24-08674-f003:**
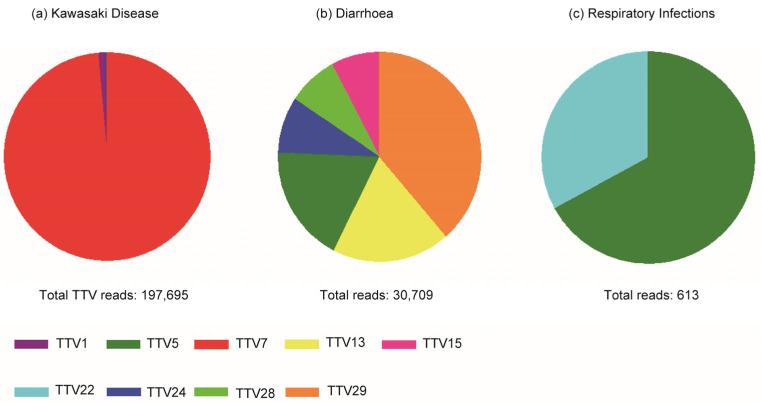
Composition of TTV species in the reads pooled from the KD, diarrhea control, and respiratory infection control groups. The reads obtained from these three groups were mapped to the reference sequences of all the TTV species.

**Figure 4 ijms-24-08674-f004:**
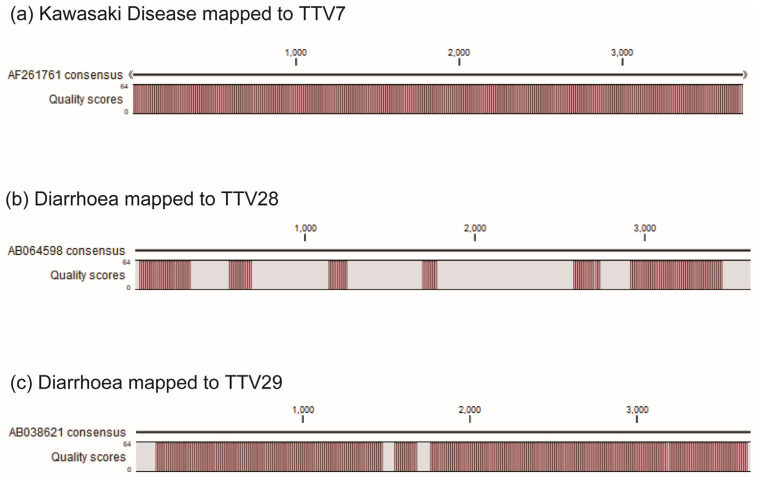
Quality scores of the consensus sequences derived from the reads mapped to the TTV reference sequences. Quality scores were estimated for consensus sequences derived from (**a**) reads in the KD group mapped to the TTV7 reference sequence, (**b**) reads in the diarrhea control group mapped to TTV29, and (**c**) reads in the diarrheal control group mapped to TTV28.

**Figure 5 ijms-24-08674-f005:**
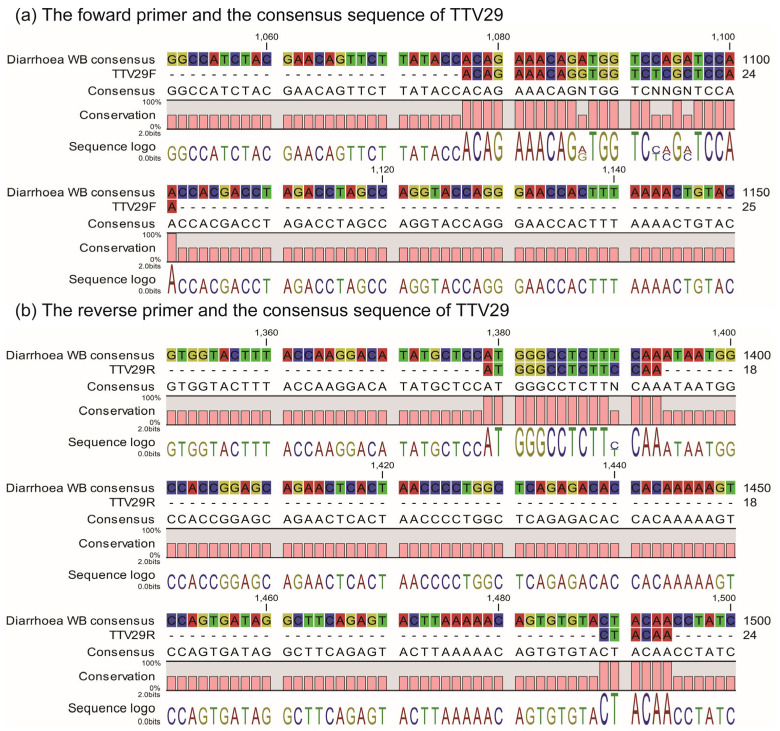
Alignment between our TTV29 PCR primer set and the TTV29 consensus sequence constructed from NGS reads. Our PCR primer set, which was designed based on the Genbank reference sequence (AB038621), was aligned with the consensus sequence constructed from the NGS reads from the diarrhea control group. Our forward primer was largely consistent with the reference sequence (**a**), whereas the reverse primer site was interrupted by a large insertion in the consensus sequence (**b**).

**Table 1 ijms-24-08674-t001:** Kawasaki disease patients and matched control individuals.

Individual	Sex	Kawasaki Disease	Diarrhea	Respiratory Infections
		Age (years)		
1	M	1.1	1.1	1.5
2	F	3.3	3.3	3.7
3	F	8.0	4.7	5.3
4	M	2.8	3.0	2.0
5	F	2.3	1.9	2.6
6	F	0.83	1.34	0.96
7	M	4.2	3.8	3.7
8	M	0.27	0.87	0.88
9	M	0.75	1.03	0.92
10	M	3.5	2.7	3.1
11	F	0.45	1.1	0.85

**Table 2 ijms-24-08674-t002:** Species-specific TTV oligonucleotides.

Primer	Sequence (5′-> 3′)	Position (nt)	Amplicon Size (bp)
TTV-1F	ACAGATCTTTGTGACATGGTGC	1285–1306	103
TTV-1R	GGAAGTTCACAACCACAGAGTC	1387–1366	
TTV-5F	TTGTGGTTATAGAGGCAACGGT	1195–1216	89
TTV-5R	AAGAACCTGGAAGTTGCAACAG	1283–1262	
TTV-7F	TCCCCCTAGTAACTATCAGTGCCT	1323–1346	113
TTV-7R	GTTATACCAGTAGGGATCTAAAATCTG	1435–1409	
TTV-13F	TCCAACTGAACTTAACCGCAGC	1224–1245	131
TTV-13R	AAATTGCGGAAGGTCTATATTGAGAC	1354–1329	
TTV-14F	TGCGACGTTAACCTTGTGCAA	1343–1363	98
TTV-14R	ACTTGGAAGGTCACACAAGGAGT	1440–1418	
TTV-15F	TTGTAACTTTGCGGTCAACTGC	1327–1348	88
TTV-15R	AACACCTGGAAGGTGGTGCAA	1414–1394	
TTV-22F	TGGTTACTGCGGCTGACTTTC	1563–1583	81
TTV-22R	CTTTCAACACTTGGAACGTTGTG	1643–1621	
TTV-24F	TTTCTGCGGCTAGCTTCATGC	1247–1267	81
TTV-24R	TGTCTTTCAACACCTGGAAGG	1327–1307	
TTV probe *	FAM/TCCGTTCKGCTCACCACWAAC/BHQ1		
TTV-28F	ACAGCCCATACTTTTTAACACCGCGA	1655–1680	722
TTV-28R	TGACACTCTTTTAAGACTTGCCGAGCT	2376–2350	
TTV-29F	ACAGAAACAGGTGGTCTCGCTCCAA	1078–1102	324
TTV-29R	CTTGTAGTTGGAAGAGGCCCATGCT	1401–1377	

* This probe was used to detect and quantify TTV species 1, 5, 7, 13, 14, 15, 22, and 24. FAM: fluorescein-labeled amidite. BHQ1: black hole quencher-1 (Sigma-Aldrich, St. Louis, MO, USA).

**Table 3 ijms-24-08674-t003:** Comparison between high-throughput sequencing of pooled samples and ssTTV-PCR of individual samples.

	Group ^†^	NGS ^‡^	ssTTV-PCR ^¶^	Comparison
TTV1	KD	+	+	Consistent
	Diarrhea	-	-	Consistent
	Respiratory	-	-	Consistent
TTV5	KD	+	+	Consistent
	Diarrhea	+	+	Consistent
	Respiratory	+	+	Consistent
TTV7	KD	+	+	Consistent
	Diarrhea	-	-	Consistent
	Respiratory	-	-	Consistent
TTV13	KD	-	-	Consistent
	Diarrhea	+	+	Consistent
	Respiratory	-	-	Consistent
TTV15	KD	+	+	Consistent
	Diarrhea	+	+	Consistent
	Respiratory	-	+	Inconsistent
TTV22	KD	-	-	Consistent
	Diarrhea	-	+	Inconsistent
	Respiratory	+	+	Consistent
TTV24	KD	+	+	Consistent
	Diarrhea	+	+	Consistent
	Respiratory	-	-	Consistent
TTV28	KD	-	-	Consistent
	Diarrhea	+	-	Inconsistent
	Respiratory	-	-	Consistent
TTV29	KD	-	+	Inconsistent
	Diarrhea	+	-	Inconsistent
	Respiratory	-	-	Consistent

^†^ Kawasaki disease, diarrhea, or respiratory infections. ^‡^ NGS was applied to the DNA samples pooled in each group. ^¶^ “+” was assigned if at least 1 of the 11 subjects in the group was positive by TTV-PCR.

**Table 4 ijms-24-08674-t004:** Statistical association between Kawasaki disease and the presence of a TTV species identified by TTV-PCR.

TTV	KD^†^	Diarrhea ^†^	Respiratory ^†^	Odds Ratio ^‡^	*p*-Value ^‡^
Any TTV	9	8	9	1.69	1.0000
TTV1	1	0	0	2.00	0.3333
TTV5	1	2	3	0.295	0.5926
TTV7	2	0	0	4.83	0.1111
TTV13	0	2	0	0.828	0.5556
TTV15	3	3	1	1.82	0.6461
TTV22	0	1	2	0.520	0.5556
TTV24	2	4	0	1.00	1.0000

^†^ Number of individuals positive for the TTV species out of the 11 individuals in the group. ^‡^ Odds ratios and *p*-values were estimated by exact logistic regression for matched samples.

## Data Availability

The accession number for the NGS data used in our analysis is DRA007000.

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
