# Peer review of "Identification of Torquetenovirus Species in Patients with Kawasaki Disease Using a Newly Developed Species-Specific PCR Method"

_ijms, 2023, doi:10.3390/ijms24108674_

Round 1

Reviewer 1 Report

The paper “Identification of Torquetenovirus (TTV) species in patients with Kawasaki disease (KD) using a newly developed species-specific PCR method” reports on a short cohort of children with KD in whom a high viral load of TTV was found. None of the control individuals were interested by the same finding. In particular, the authors developed a species-specific PCR methodology that differentiates all TTV species. The paper found that TTV7 showed a marginal correlation with KD.

In the Discussion the authors state that although it has been widely assumed that KD has an infectious origin, the aetiological agent has yet to be identified. During the global outbreak of COVID-19 the SARS-CoV-2 was reported to cause a paediatric illness clinically similar to KD. To tell the truth a host of reports have emphasized the potential role of viruses in triggering inflammation which generates KD in genetically predisposed children even before the COVID-19 pandemic. However, it is correct that the univocal aetiology of KD has not yet been defined (as written in the Introduction).

I would add some ‘conclusive remarks’ at the end of the paper, highlighting that this study is preliminary and that no definite conclusions could be drawn by such findings.

The discussion should also have a wider breath, for instance, writing that vascular complications for KD are related to the “extent “of inflammation, as shown by different papers, and that many clues to predict non-responsiveness to KD-treatment have been analyzed or validated in non-European countries (please, cite the review by Rigante et al which just appeared in Int J Mol Sci 2016;17:278). The relationship of clinical manifestations, labwork clues and different viral aetiology of KD is a future issue of study with the goal of improving the knowledge of this complex disorder and vascular prognosis of KD patients.

Author Response

Dear Dr Letrari Liu,

We appreciated that you found this excellent and kind Reviewer whose valuable advices improved the quality and readability  of our manuscript. We would like to respond to the individual comments from this Reviewer as follows:

Comment: The paper “Identification of Torquetenovirus (TTV) species in patients with Kawasaki disease (KD) using a newly developed species-specific PCR method” reports on a short cohort of children with KD in whom a high viral load of TTV was found. None of the control individuals were interested by the same finding. In particular, the authors developed a species-specific PCR methodology that differentiates all TTV species. The paper found that TTV7 showed a marginal correlation with KD.

Response: We are grateful to this Reviewer who read this manuscript and our previous article very carefully.

Comment: In the Discussion the authors state that although it has been widely assumed that KD has an infectious origin, the aetiological agent has yet to be identified. During the global outbreak of COVID-19 the SARS-CoV-2 was reported to cause a paediatric illness clinically similar to KD. To tell the truth a host of reports have emphasized the potential role of viruses in triggering inflammation which generates KD in genetically predisposed children even before the COVID-19 pandemic. However, it is correct that the univocal aetiology of KD has not yet been defined (as written in the Introduction). I would add some ‘conclusive remarks’ at the end of the paper, highlighting that this study is preliminary and that no definite conclusions could be drawn by such findings.

Response: Following this useful suggestion, we added a conclusive paragraph at the end of Discussion as in (Page 13 lines 31 – 32), as in:

  1. Conclusions.

We evaluated whether ssTTV-PCR can be used to explore the relationship between TTV species and KD. We did not intend to give a definitive answer to the aetiological question of KD. Instead, this preliminary study explored the strength and weakness of this newly developed tool. ssTTV-PCR is more sensitive and less costly than NGS. However, due to the mutations in the primer sites, ssTTV-PCR may fail to identify an evolving TTV species. Therefore, a future study which examines the role of TTV in the pathogenesis of KD should combine the benefit of the PCR based methodology and that of NGS.

Comment: The discussion should also have a wider breath, for instance, writing that vascular complications for KD are related to the “extent “of inflammation, as shown by different papers, and that many clues to predict non-responsiveness to KD-treatment have been analyzed or validated in non-European countries (please, cite the review by Rigante et al which just appeared in Int J Mol Sci 2016;17:278). The relationship of clinical manifestations, labwork clues and different viral aetiology of KD is a future issue of study with the goal of improving the knowledge of this complex disorder and vascular prognosis of KD patients.

Response: We completely agreed with this useful opinion, and hence inserted the following statement in the Discussion (Page 11 line 24-25):

A future study should examine the relationship not only between TTV and development of KD. The association between TTV and clinical outcome of KD, most importantly cardiac complications, should also be an important research topic. Meanwhile, development of cardiac lesions is closely associated with refractoriness to IVIG. However, the scoring models which were developed to predict IVIG-resistance in Japan are not equally sensitive or specific in non-Japanese populations [51]. This fact implies that relationship between complications of KD and the possible causative agents including TTV may also be dependent on the study population. Therefore, further efforts should be continued to explore the aetiological agent(s) of KD, in diverse ethnicities, using up-to-date technologies including ssTTV-PCR.

In this paragraph, we cited the following important article (Page 17 line 50):

  1. Rigante, D.; Andreozzi, L.; Fastiggi, M.; Bracci, B.; Natale, M. F.; Esposito, S., Critical Overview of the Risk Scoring Systems to Predict Non-Responsiveness to Intravenous Immunoglobulin in Kawasaki Syndrome. Int J Mol Sci 2016, 17, (3), 278.

Thank you very much for all these extremely helpful comments.

Reviewer 2 Report

Thank you very much for the opportunity to review the manuscript entitled “Identification of Torquetenovirus species in patients with Kawasaki Disease using a newly developed species-specific PCR method”.

The manuscript provides a new technique for lab detection of torquetenovirus, a recently described virus with many potential implications. It describes a cheaper and potentially more sensitive technique than NGS, but studied in a very short sample size.

As a major comment, I’d repeat the analysis increasing sample size (maybe using healthy controls paired also by sex and age) before suggesting larger studies. 

I also would like to know more details about respiratory infection controls (viral agent, severity of cases…). It would be more interesting if controls had other conditions similar to KD (SGA, toxic shock, adenovirus).

Thank you, very interesting work but I'd improved the content to be considered for this length of paper (maybe try with as a short communication).

Good enough.

Author Response

Dear Dr Letrari Liu,

We are grateful to you for assigning this expertised Reviewer whose comments helped us improve the quality of our manuscript. We understand that finding such an excellent reviewer is increasingly difficult now that thousand of journals are competing for a limited number of qualified reviewers. We are pleased to respond to the individual comments as follows:

Comment: Thank you very much for the opportunity to review the manuscript entitled “Identification of Torquetenovirus species in patients with Kawasaki Disease using a newly developed species-specific PCR method”. The manuscript provides a new technique for lab detection of torquetenovirus, a recently described virus with many potential implications. It describes a cheaper and potentially more sensitive technique than NGS, but studied in a very short sample size.

Response: We appreciated that this Reviewer has read our manuscript very carefully and summarized it comprehensively and beautifully.

Comment: As a major comment, I’d repeat the analysis increasing sample size (maybe using healthy controls paired also by sex and age) before suggesting larger studies. 

Response: We agreed with this Reviewer that the sample size should ideally be increased. Unfortunately, however, this prospective research project was approved by the Ethics committee of Japan Community Health Care Organisation Osaka Hospital only for the duration specified in ref [16] and Page 12 line 26. As this Reviewer suggested, we conducted another research project with a greater sample size. However, this larger project was approved in other hospitals. The result from the latter project will be submitted elsewhere and we are not allowed to disclose it at this time of moment.

Comment: I also would like to know more details about respiratory infection controls (viral agent, severity of cases…). I

Response. Following this important suggestion, we added the following paragraph (Page 3 line 74-82):

2.2 High-throughput NGS sequencing and bioinformatical analysis.

Livermore Metagenomics Analysis Tool Kit was applied to the high-throughput NGA datasets as described previously [16]. Briefly, a large number of viral reads were identified in the WB DNA which was pooled from 11 KD patients, and 99% of these reads were TTV7. In contrast, only a very small number of viral reads were found in the WB pooled from 11 diarrheal patients or from 11 children with respiratory infections. Similarly, 76% of the viral reads in the NPA pooled from 11 KD patients were TTV7. Rhinovirus A, respiratory syncytial virus and human metapneumo virus were the most abundant viral species in the NPA obtained from 11 children with respiratory infections.

Comment: it would be more interesting if controls had other conditions similar to KD (SGA, toxic shock, adenovirus).

Response: We found valuable this suggestion to compare KD and KD-mimicking illnesses (e.g. toxic shock syndrome, adenovirus infections). However, per study protocol, we did not enroll a child with symptoms mimicking KD as in (Page 12 line 26):

A child with mucocutaneous symptom(s) which resembles KD (e.g. rash, conjunctivitis) was not enrolled as a control subject because he/she may have incomplete KD.

Comment: Thank you, very interesting work but I'd improved the content to be considered for this length of paper (maybe try with as a short communication).

We are grateful to this Reviewer whose insightful advices greatly helped us correct and improve our manuscript. Thank you very much.
